

# Persistence of upper stratospheric winter time tracer variability into the Arctic spring and summer

David E. Siskind[1], Gerald E. Nedoluha[2], Fabrizio Sassi[1], Pingping Rong[3], Scott M. Bailey[4], Mark E. Hervig[5], and Cora E. Randall[6]

[1]Space Science Division, Naval Research Laboratory, Washington DC
[2]Remote Sensing Division, Naval Research Laboratory, Washington DC
[3]Center for Atmospheric Sciences, Hampton University, Hampton VA
[4]Bradley Department of Electrical and Computer Engineering, Virginia Tech, Blacksburg, VA
[5]GATS-Inc., Driggs, ID
[6]Laboratory of Atmospheric and Space Physics and Department of Atmospheric and Oceanic Sciences, University of Colorado, Boulder CO

*Correspondence to:* David Siskind (david.siskind@nrl.navy.mil)

**Abstract.** Using data from the Aeronomy of Ice in the Mesosphere (AIM) and the Aura satellites, we have categorized the interannual variability of winter and spring time upper stratospheric $CH_4$. We further show the effects of this variability on the chemistry of the upper stratosphere throughout the following summer. Years with strong mesospheric descent followed by dynamically quiet

springs, such as 2009, lead to the lowest summertime $CH_4$. Years with relatively weak descent, but strong springtime planetary wave activity, such as 2011, have the highest summertime $CH_4$. By sampling the Aura Microwave Limb Sounder according to the occultation pattern of the AIM Solar Occultation for Ice Experiment, we show that summertime upper stratospheric ClO almost perfectly anticorrelates with the $CH_4$. This is consistent with the reaction of atomic chlorine with $CH_4$ to form

the reservoir species, HCl. The summertime ClO for years with strong, uninterrupted mesospheric descent is about 50% greater than in years with strong horizontal transport and mixing of high $CH_4$ air from lower latitudes. Small, but persistent effects on ozone are also seen such that between 1-2 hPa, ozone is about 4-5% higher in summer for the years with the highest $CH_4$ relative to the lowest. This is consistent with the role of the chlorine catalytic cycle on ozone. These dependencies may of-

fer a means to monitor dynamical effects on the high latitude upper stratosphere using summertime ClO measurements as a proxy. Also, these chlorine controlled ozone decreases, which are seen to maximize after years with strong uninterrupted wintertime descent, represent a new mechanism by which mesospheric descent can affect polar ozone. Finally, given that the effects on ozone appear to persist much of the rest of the year, the consideration of winter/spring dynamical variability may

also be relevant in studies of ozone trends.



## 1 Introduction

There has recently been great interest in the variability of middle atmospheric trace constituents at high latitudes in the late winter and early spring. This interest has been fueled, in part, by the occurrence of prolonged sudden stratospheric warmings (SSWs) which can perturb the composition and structure of the stratosphere and mesosphere for many weeks (Manney et al., 2008a, b,2009). These so-called extended SSWs are characterized by elevated stratopauses which reform near and above 80 km (Siskind et al., 2007; Manney et al., 2009). During the recovery phase of these extended events, the anomalous zonal wind flow alters the gravity wave propagation to the mesosphere, thus perturbing the mean meridional circulation and driving a dramatic descent of mesospheric air down to the stratosphere. For example, Bailey et al. (2014) have shown that mesospheric air enhanced in nitric oxide and depleted in water vapor and $CH_4$ can descend from near 90 km in early February down to 40 km by early April. Bailey et al. (2014) focused on the 2013 SSW; other analogous events occurred in 2004, 2006 and 2009 (Manney et al., 2005, 2009; Randall et al., 2009). An additional motivation for much of the above studies is the interest in quantifying the extent to which the enhanced nitric oxide can cause reductions in polar upper stratospheric ozone (Funke et al., 2014).

There has been less attention paid to what happens to these dramatic perturbations as the spring progresses and the wintertime circulation transitions into a summer pattern. It has long been recognized that the winter to spring transition is characterized by a decay and breakdown of the winter time westerly jet and its eventual replacement by a zonal mean easterly flow around the polar region. This is known as the stratospheric final warming (SFW) (Hu et al., 2014). It has been observed that certain remnants of wintertime dynamical (Hess, 1991) or chemical tracer features (Orsolini, 2001; Lahoz et al., 2007) can persist well into the summer season. Most recently, work has focused upon specific events whereby the SFW can occur rather abruptly with a significant late season planetary wave event (Allen al., 2011; Siskind et al., 2015a; Fiedler et al., 2014). These planetary waves can transport low latitude anticyclonic air poleward. This air can displace the winter polar vortex and then remain "frozen in" for a period of weeks or longer in late spring and early summer (Manney et al. , 2006). Alternatively, this transition can occur gradually without significant wave activity. In the former case, the upper mesosphere often experiences cooler and wetter conditions which can lead to the early onset of the polar mesospheric cloud (PMC) season. In the latter case, the upper mesosphere remains warmer and drier. Siskind et al. (2015a) showed that 2011 and 2013 were years with an abrupt winter-to-spring transition and 2008 was a spring with negligible planetary wave activity. They used these years to define the extremes in spring time planetary wave activity and associated temperatures.

From the above, we can define four general scenarios for the transition from winter to summer based upon the combination of the two perturbations outlined above. We can have a year with extended descent of mesospheric air (typically the result of a extended SSW) or a winter with weak descent. These winters can be followed by springs with either an abrupt planetary wave transition





to a summer circulation or with a slower gradual transition. The purpose of this paper is to catego-
rize the four possible combinations of these springtime scenarios and how they are manifested in
the variability of trace constituents such as $CH_4$, ClO and ozone. Among our results, we will show
that under certain circumstances, the zonal mean distribution of these trace constituents can be per-
turbed for many months even into the autumn. This is important because while the summer upper
stratosphere is generally understood to be under radiative and photochemical control (Andrews et
al., 1987), we will show how the zonal mean composition can be sensitive to dynamical changes that
might have occurred over half a year prior.

## 2 Observations and Model

### 2.1 SOFIE and MLS data

Our primary data come from the Solar Occultation for Ice Experiment (SOFIE) (Gordley et al. ,
2009) on the Aeronomy of Ice in the Mesosphere (AIM) satellite (Russell et al., 2009) and the
Microwave Limb Sounder (MLS) (Santee et al., 2008; Froidevaux et al., 2008) on the Aura satellite
(Waters et al. , 2006). SOFIE measures profiles of temperature, aerosols (ice and meteoric smoke)
and $O_3$, $H_2O$, $CO_2$, $CH_4$ and NO using the solar occultation technique. Since the AIM satellite is in a
sun-synchronous polar orbit, the latitude of the occultations approximately tracks the terminator and
is above 82° near equinox and near 65° at solstices. The vertical resolution is about 2 km. This work
uses version 1.3 SOFIE data. SOFIE $CH_4$ data has previously been presented by Bailey et al. (2014)
and Siskind et al. (2015b); ongoing validation studies with the Atmospheric Chemistry Experiment
suggest general agreement to 12%. Here we emphasize the relative year to year variations.

Like AIM, the Aura satellite is also in a sun-synchronous orbit. However, unlike SOFIE, because
MLS observes ClO and $O_3$ in emission, data is obtained over all latitudes up to about 82°N. We used
Version 4.2 data. The MLS ozone was validated by Froidevaux et al. (2008) and used in a study of
lower mesospheric photochemistry by Siskind et al. (2013). The ClO data has been validated by
Santee et al. (2008) and compared with groundbased data by Nedoluha et al. (2011). Santee et al.
(2008) show that the precision of the MLS ClO decreases for pressures less than 2 hPa; however,
since we only show monthly averages, this is not a problem for the present study. It is also common
practice to subtract the nighttime data from the daytime data (Santee et al., 2008; Nedoluha et al.,
2008) in order to reduce systematic biases; however, for the high latitude spring/summer conditions
shown here, there are often no night periods. Thus a given monthly average was constructed using
data from all local times without any background subtraction. The vertical resolution of the MLS
ClO observation (3-4 km) is somewhat coarser than SOFIE. We thus interpolated the SOFIE profile
to the MLS grid.




### 2.2 The Whole Atmosphere Community Climate Model (WACCM)

We also compare some of our results with WACCM [Garcia et al., 2007]. WACCM is the high altitude atmospheric component of the NCAR Community Earth System Model version 1 (CESM1). In its standard configuration, WACCM has 66 vertical levels from the ground to about $5.9 \times 10^{-6}$

95 hPa ( 140 km geometric height) and a horizontal resolution of $1.9°$ latitude x $2.5°$ longitude. See Garcia et al. (2007) for a detailed discussion of the model climate and parameterizations. This version of WACCM uses specified dynamics (SD) provided by the Navy Operational Global Atmospheric Prediction System- Advanced Level Physics High Altitude (NOGAPS-ALPHA) (Marsh, 2011; Sassi et al., 2013). NOGAPS-ALPHA is the high altitude extension of the then operational

Navy's weather forecast system up to about 90-92 km (Eckermann et al., 2009). Siskind et al. (2015b) have already shown that the combination of WACCM and NOGAPS-ALPHA (hereinafter called WACCM/NOGAPS) produced a successful representation of the descent of enhanced upper mesospheric and lower thermospheric nitric oxide (NO) and depleted $CH_4$ into the upper stratosphere/lower mesosphere. By contrast, WACCM nudged by MERRA did not (see also Randall et al.,

2015). Since mesospheric descent is so important for understanding our present results, we only use WACCM/NOGAPS for this study. Unfortunately, of the seven years considered here (2008-2014), WACCM/NOGAPS is only available for the first two. We thus can not use it to reproduce all the variability seen in the SOFIE data. However, by comparing summer results from 2009 with 2008, we can provide a broader context to the latitudinal extent of the $CH_4$ changes and their effect on the

chlorine and ozone chemistry of the upper stratosphere.

## 3 Results

### 3.0.1 $CH_4$

Our specific interest is to highlight the consequences of the variations in upper stratospheric $CH_4$ as observed by SOFIE and shown in Figures 1 and 2. These figures illustrate the great variability that

occurs in $CH_4$ each winter and spring. Figure 1 shows that each year is characterized by the descent of low values of $CH_4$ from the mesosphere in the period from February to early April (roughly Day 30 to Day 110). This descent is characterized by large interannual variability and was strongest in 2009 and 2013. These were years with prolonged SSWs followed by elevated stratopauses and have been covered in the literature (Manney et al., 2009; Randall et al., 2009; Bailey et al., 2014). The

difference between 2009 and 2013 is that in 2013, there was a large frozen in anticylonce event (FrIAC; Manney et al., 2006) that transported high values of $CH_4$ to high latitudes (Siskind et al., 2015a) whereas in 2009, no such spring time disturbance was evident. This is clearly seen in Figure 2 where the $CH_4$ jumps from below 0.1 ppmv on Day 100 to over 0.3 ppmv by Day 120. Years with a more moderate and shorter period of winter/early spring descent are 2010 and 2012. These





two years did not have elevated stratopause events as in 2009 and 2013, but there were wintertime
SSWs in both years and Straub et al. (2012) discussed the descent of dry air at high latitudes in
the lower mesosphere during the late winter of 2010. The springtime vortex breakdown occurred
relatively gradually over many weeks in March and April for both 2010 and 2012 and thus there was
no transport of high $CH_4$ in either spring. These years ended up being close to 2009 in having low
values of $CH_4$ persist into the summer. Even less mesospheric descent was seen in 2008 and the least
descent was seen in 2011 and 2014. 2011 was characterized by a strong undisturbed stratospheric
polar vortex (Manney et al., 2011). Then in early April (Day 95) of that year, the largest FrIAC of
the 36-year Modern Era Retrospective Analysis for Research and Applications (MERRA) dataset
was recorded (Allen et al., 2011; Thieblemont et al., 2013), causing a significant jump in upper
stratospheric $CH_4$.

After the spring, there is a 2nd period of decreasing $CH_4$ in the summer (most noticeable after Day
200). This summer time decrease is due to photochemistry (Funke et al., 2014) as the production of
$O(^1D)$ and OH, both of which oxidize $CH_4$, peak at high summer latitudes in the upper stratosphere
(LeTexier et al., 1988). Since the upper stratosphere at this time of year is dynamically quiet, the
year to year variability in summer $CH_4$ is driven by the winter and springtime dynamics. This can
be seen in Figure 2, which compares time series of upper stratospheric $CH_4$ for the 6 years shown in
Figure 1 plus 2014. The figure shows that the lowest summer $CH_4$ was generally in 2009; this is the
direct consequence of the late winter descent that persisted without interruption until early April. By
contrast, the highest summer $CH_4$ was in 2011 which is the result of the dynamically quiet winter
followed by the FrIAC in early April that caused the $CH_4$ to almost double. The other 5 years are
intermediate, although as noted above, 2010 and 2012 are close to 2009. For all seven years, once
the relative abundance of $CH_4$ was established by May 1st (Day 121), it remained mostly unchanged
until October (around Day 280).

Table 1 presents an idealized categorization of how the summer level of Arctic upper stratospheric
$CH_4$ can be placed in the context of the four categories of wintertime descent and early spring
dynamical variability. The years 2008, 2009, 2011, 2013 are most representative of these idealized
cases. The other years are more intermediate; as noted above, 2010 and 2012 were closer to 2009
in having relatively strong late winter descent and a relative absence of spring time wave activity.
2014 is closer to 2011. As seen in Figure 2, there was a 50% increase in $CH_4$ in late March 2014
and we have previously, tentatively suggested that there was a FrIAC event in that spring (Siskind et
al., 2014).

### 3.0.2 ClO

Here we explore the chemical consequences of the $CH_4$ variations illustrated above. $CH_4$ has long
been known to play an important role in partitioning stratospheric chlorine (Solomon and Garcia,
1984). Specifically, the reaction $Cl + CH_4 \rightarrow HCl + CH_3$ means that active chlorine ($ClO_x = Cl +$



ClO) should vary inversely with CH$_4$. For example, Siskind et al. (1998) documented an increase in ClO during the early years of the Upper Atmospheric Research Satellite (UARS) mission which was explained as a direct consequence of the decrease in CH$_4$ observed by Nedoluha et al. (1998). Froidevaux et al. (2000) observed a general anticorrelation between variations in ClO and CH$_4$ in the tropics.

Figure 3 shows that this anticorrelation also exists between high latitude CH$_4$ and ClO during the spring and summer. It plots monthly averaged SOFIE CH$_4$ against MLS ClO (sampled at the SOFIE occultation latitudes) for the period May-August. Note there is a general increase in ClO from late spring to late summer. This is consistent with the seasonal decrease in CH$_4$ and was discussed by Considine et al. (1998). Concerning the year-to-year variability, the highest summertime ClO for the seven year period is in 2009. This is a legacy of the strong uninterrupted descent which followed the January 2009 SSW. Other years with relatively high ClO include 2010 and 2012 which, as we have discussed, were also years similar to 2009 in their combination of winter descent and spring planetary waves. The lowest summertime ClO is in 2011. This is the result of the strong FrIAC event which occurred in April 2011. The general range of summer ClO which stems from the above winter/spring dynamical variability is about 50%.

To get a broader picture of the ClO and CH$_4$ changes at latitudes other than the narrow range sampled by SOFIE, Figure 4 shows the monthly average zonal mean WACCM/NOGAPS ClO and CH$_4$ difference fields for Aug 2009 minus Aug 2008. Also shown in the right hand plots are profiles that are compared with MLS (for ClO) and SOFIE (for CH$_4$) for the SOFIE occultation latitude (given by the dashed white line in the color panel). The comparison between the model and the data is excellent. Since the difference between 2009 and 2008 represents about half the difference between the extreme years discussed above (2009 and 2011), one can multiply the ClO and CH$_4$ difference values in Figure 4 by a factor of two to get an estimate of the full range. The model shows that the low 2009 CH$_4$ and high 2009 ClO shown in Figure 4 are part of a broad region of perturbation extending from 40-50°N to the pole and covering the altitude region between about 1 and 8 hPa. There may be a small vertical offset, perhaps one grid point, whereby the model profile is shifted slightly downward relative to both the MLS and SOFIE data. A similar offset was recently noted by Siskind et al. (2015b) in their WACCM/NOGAPS simulation of the 2009 descent of mesospheric NO$_x$. Since the summer CH$_4$ depletion is a consequence of the winter descent, this offset may reflect the small discrepancy seen by Siskind et al. (2015b).

Figure 4 shows that the effect of the CH$_4$ on ClO occurs over a relatively deep layer in the upper stratosphere; the detailed plots of the time behavior of CH$_4$ and ClO, specifically Figures 2 and 3, represent only the uppermost edge of this larger perturbation. The reason for focusing on this narrower region is that these altitudes, between 1-3 hPa, are where the chlorine cycle is affecting the ozone. This is discussed in the next section.




### 3.0.3 Ozone

Figure 5 presents a time series of upper stratospheric ozone in a format similar to Figure 2 for $CH_4$. Only 4 years are shown because in summer, the curves almost overlap and it would be hard to distinguish all 7 years clearly. The 4 years shown correspond to the representative years given in Table 1. The figure shows very large variability in March and April, both intra- and inter-annually. This is largely driven by the large temperature variability, which itself is dynamically driven, as discussed by several authors (Siskind et al., 2015a; McCormack et al., 2006; Smith, 1995; Froidevaux et al., 1989). Of interest here is that after May 1st the interannual variability becomes very small, but is not zero. Also it shows that the relative abundance from year to year remains generally fixed throughout the summer into the autumn. This small remaining difference is due to chlorine chemistry as seen below.

Figure 6 shows the zonal and monthly averaged ozone loss rates from the $HO_x$, $ClO_x$ and $NO_x$ catalytic cycles for June 2008 and 2009 at 80°N calculated by WACCM/NOGAPS. The expressions for these terms are from McCormack et al., (2006). The figure shows that the chlorine loss is about 20% larger in 2009 than in 2008 and that this is centered in a narrow layer from 1-3 hPa. The HOx cycle shows little change, but the $NO_x$ cycle actually shows the opposite effect, i.e. decreased loss in 2009. The net effect is that in the 1-2 hPa layer, the overall ozone loss is about 2% greater in 2009. Between 3-7 hPa, there is a small decrease in ozone loss in 2009. These changes agree well with observed ozone changes as seen by MLS. This is shown in Figure 7 which presents an altitude profile of the ozone change from WACCM/NOGAPS compared with MLS. The figure shows the relative 2009 ozone decrease near 1-2 hPa, corresponding to the increase in chlorine loss. The model slightly underestimates this compared with MLS; this may be consistent with the small underestimate of the chlorine enhancement that we discussed in Figure 4 above. From 4-6 hPa, there is a small ozone increase in 2009 which corresponds to the small reduction in $NO_x$ loss seen in Figure 6.

Figure 8 shows that the ozone change over the entire seven year period is consistent with the above analysis for 2008 and 2009. Figure 8 presents monthly averaged correlation coefficients between MLS ozone and MLS ClO (Figure 8a) and between MLS ozone and SOFIE $CH_4$ (Figure 8b) for 1.4hPa. Figure 8a shows that the approximate 5% spread in ozone values is almost perfectly anticorrelated with the 50% ClO changes shown in Figure 3 . Further, since we have previously shown that the summer ClO in the upper stratosphere reflects the interannual variability in $CH_4$, it is no surprise that MLS $O_3$, sampled at SOFIE latitudes, should almost perfectly correlate with SOFIE $CH_4$. This is shown in Figure 8b.

Finally, Figure 9 plots the linear correlation coefficient of $CH_4$ and $O_3$ as a function of altitude. Four curves are shown, corresponding to the 4 monthly averages presented in Figure 5. The figure shows that the correlation maximizes in the 1-2 hPa region with values near and above 0.9. This is to be expected from the chlorine cycle as shown in Figure 6 above. Below 2-3 hPa, the $NO_x$ cycle becomes more dominant and the link to $CH_4$ disappears. Thus the effects of uninterrupted





wintertime descent of mesospheric air on ozone may fall into two categories, separated by altitude. From 1-2 hPa the ozone reductions result from chlorine enhancements; for higher pressures, the potential for $NO_x$ enhancements dominates. We should stress however, that for 2009, there is no evidence from either our WACCM/NOGAPS simulations or from SOFIE (cf. Siskind et al., 2015b) for any enhancement of $NO_x$ at these higher pressures that might have come from the descent of mesospheric air that would be enriched in NO. Salmi et al. (2011) came to this same conclusion in their study of data from the Atmospheric Chemistry Experiment Fourier Transform Spectrometer.

## 4 Conclusions

We've shown how the chemical composition in the summertime upper stratosphere depends upon dynamical activity from the previous winter and spring. Our main result is to identify a new mechanism for summertime ClO and $O_3$ variability, namely due to $CH_4$ variations which, in turn, depend upon both the magnitude of winter time mesospheric descent and spring time planetary waves. In 2009, prolonged mesospheric descent and a relative absence of spring time wave activity lead to relatively low values of $CH_4$ which persisted throughout the summer. At the other extreme, in 2011, the lack of strong winter descent combined with an intense frozen-in-anticylcone event in early April led to $CH_4$ values which were more than twice that in 2009.

The excellent anticorrelation between MLS ClO and SOFIE $CH_4$ both validates our understanding of reactive chlorine partitioning and also offers a framework for interpreting future observations. Due to orbital precession, the latitudes of the SOFIE occultations have drifted away from polar region and SOFIE is presently unable to monitor wintertime tracer descent. However, based upon the results in this paper, perhaps MLS ClO data can be used as a proxy for this. It would also be interesting to consider whether these variations in ClO have any impact on $O_3$ trend assessments. Both the strong winter descent and the spring FrIAC phenomenon seem to be more common in recent years (Allen et al. , 2011; Manney et al., 2005). In principle, the enhanced variability we've shown here might have to be considered, at least for trend studies at high latitudes. Recent estimates of ClO trends (Jones et al. , 2011) have only considered the tropics.

Our work shows that these $CH_4$ and ClO variations have caused up to a 5% variation in upper stratospheric ozone throughout the summer and early fall. This confirms the general role of chlorine chemistry in upper stratospheric ozone. This also represents a second mechanism, in addition to that associated with descent of enhanced mesospheric $NO_x$, by which descent of mesospheric air can cause ozone reductions. Studies of spring and summer time ozone loss following strong descent years should take care to distinguish between these two mechanisms. One way to distinguish them may be according to altitude. Thus ozone decreases for p < 3 hPa (z > 40 km) are more likely the result of low $CH_4$ whereas for p > 3 hPa (z < 40 km), $NO_x$ enhancements would dominate. A likely example of this second case is shown in Figure 1 of Randall et al. (2005).





Finally, the question of whether this variability would influence trend analyses may be worth considering. There was earlier work using Upper Atmospheric Research Satellite data to look at hemispheric differences in ozone trends (Considine et al. , 1998); in light of the more recent dynamical variability seen in the NH, and its now-documented impact on ozone, perhaps this should be revisited.

*Acknowledgements.* We acknowledge the Aeronomy of Ice in the Mesosphere explorer program from the NASA Small Explorer Program. Two of us (FS and GEN) additionally acknowledge funding from the Chief of Naval Research.



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



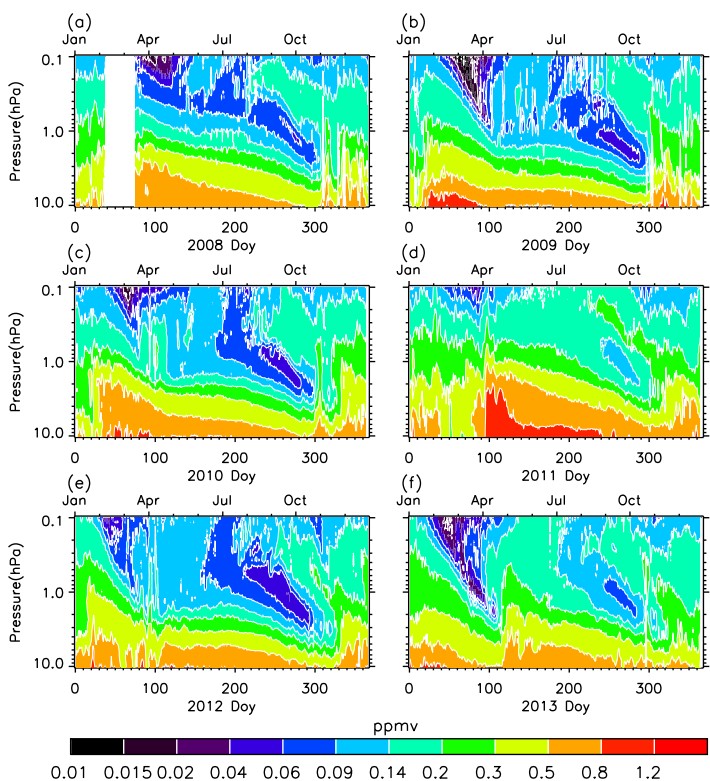

**Figure 1.** Overview of upper stratospheric and lower mesospheric zonal mean CH$_4$ observed by SOFIE for the indicated years. SOFIE observes at only 1 latitude per day in each hemisphere. This latitude varies has some variation from year to year, but is typically near 82° at the equinoxes and near 65-66° at the solstices.

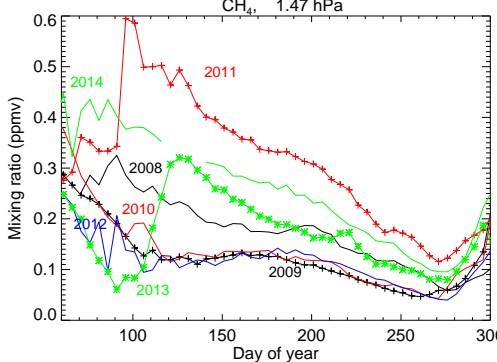

**Figure 2.** Comparison of time series of zonal mean SOFIE CH$_4$ mixing ratio for the indicated years at 1.47 hPa. The data have been grouped in 5-day bins. See Figure 1 for a discussion of the latitudes.





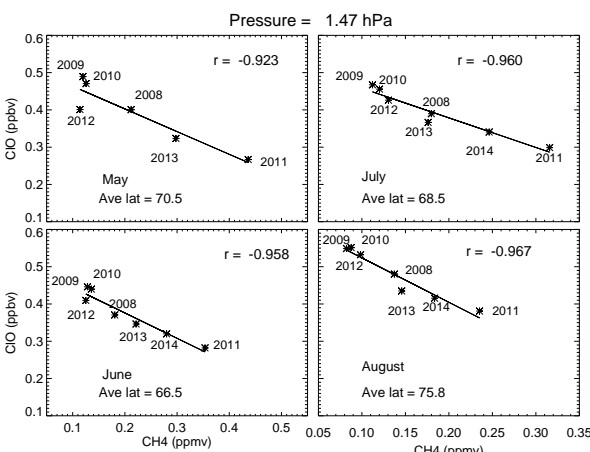

**Figure 3.** Scatterplot of zonal mean, monthly averaged MLS ClO versus SOFIE CH$_4$ at 1.47 hPa. The MLS

data are sampled at the SOFIE occultation latitude, the monthly averages of which are indicated in each panel.

The linear correlation coefficients between each dataset for each month are given in the upper right of each

panel.





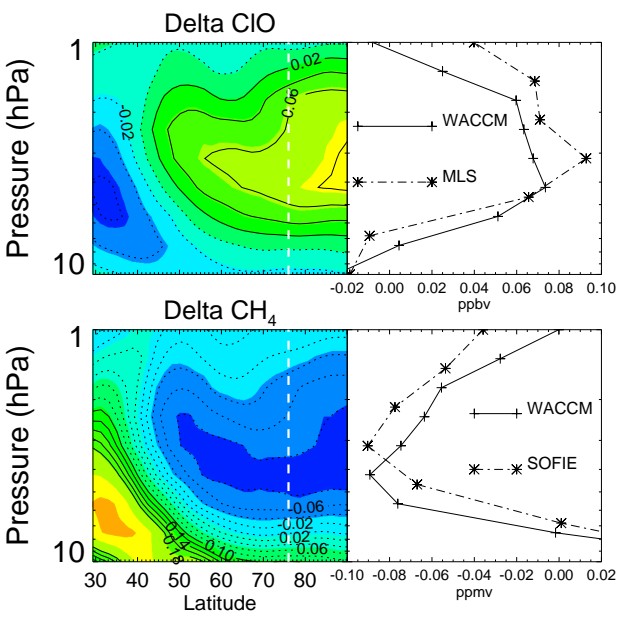

**Figure 4.** The color contours on the left are zonal mean WACCM/NOGAPS difference fields for August 2009 minus August 2008 for ClO (top) and CH$_4$ (bottom). The vertical dashed white line is the mean latitude of the SOFIE occultations for August. On the right, a vertical profile of the model difference at the SOFIE occultation latitude (solid line with plus symbols) is compared with MLS ClO and SOFIE CH$_4$ (data are dot-dashed curves with stars). Note that x-axis for the right panels are reversed from one another since the ClO change is positive while the CH$_4$ change is negative.





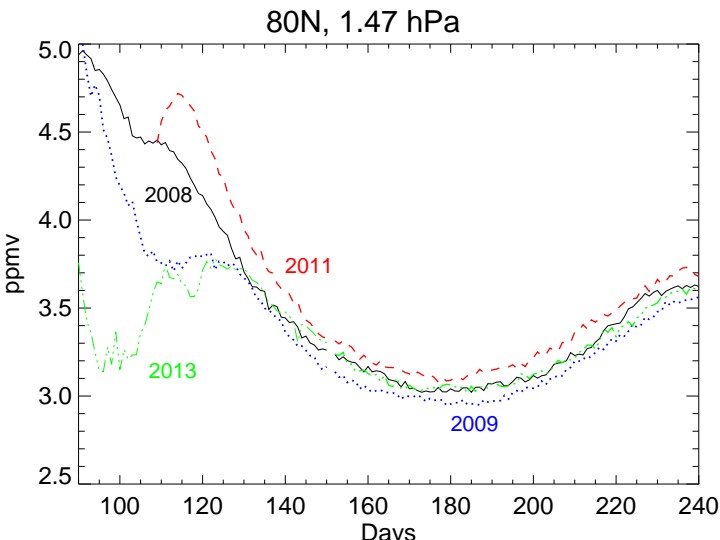

**Figure 5.** Time series of zonally averaged ozone from MLS at 80N.

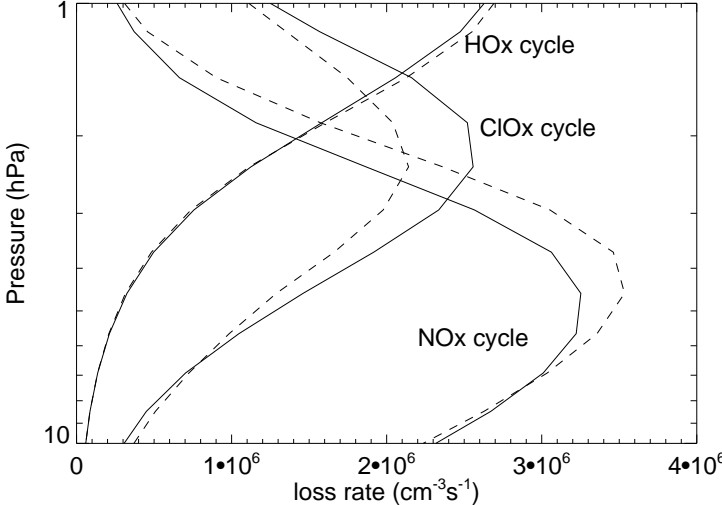

**Figure 6.** Altitude profiles of monthly and daily averaged ozone loss rates from WACCM/NOGAPS for 2009
(solid) and 2008 (dashed) at 80°N.





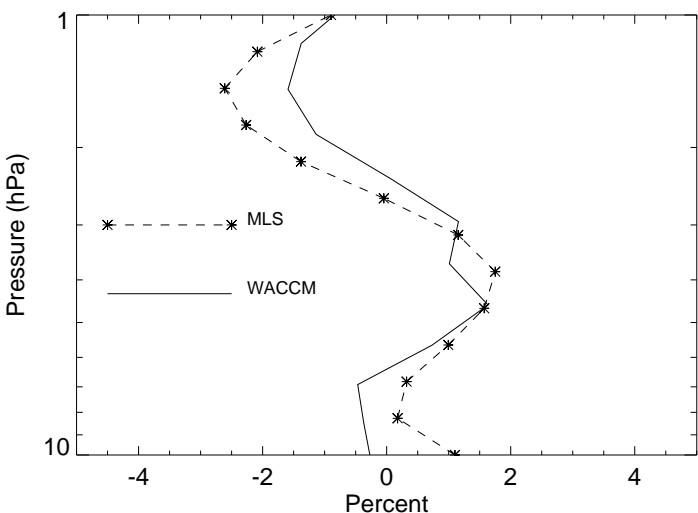

**Figure 7.** Percent change in monthly and daily averaged ozone from 2009 minus 2008 at 80°N. The solid line is from WACCM/NOGAPS and the dashed line with stars is from MLS data.

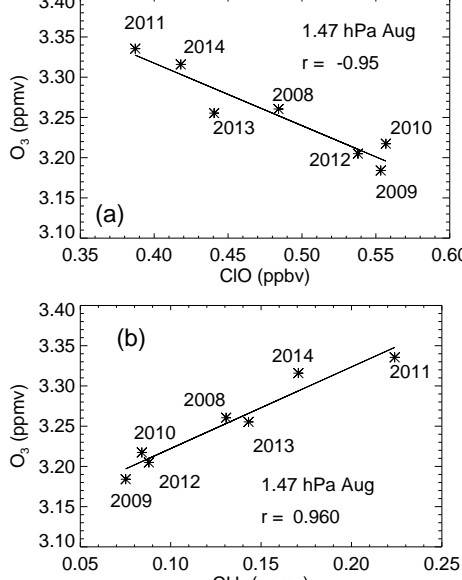

**Figure 8.** (a) Scatter plot of August monthly mean MLS $O_3$ vs. (a) MLS ClO and (b) SOFIE $CH_4$ at 1.47 hPa. The latitudes are near 78°N, corresponding to the latitude of the SOFIE occultations in August.



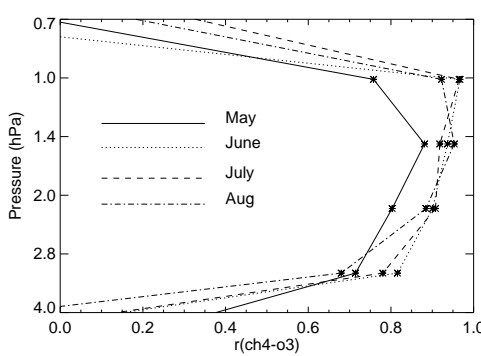

**Figure 9.** Altitude profiles of linear correlation coefficients for SOFIE $CH_4$ and MLS $O_3$ (sampled at the SOFIE occultation latitudes). The four curves are taken from zonal mean averages for May (solid), June (long dashes), July (dotted) and August (dot-dashed).

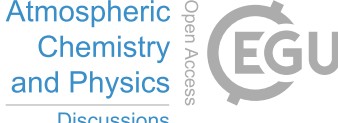



**Table 1.** Categorization of Summer Upper Stratospheric CH$_4$

| Category | Winter Descent | Spring PW | CH$_4$ value | Representative year |
|---|---|---|---|---|
| 1. | high | low | lowest | 2009 |
| 2. | high | high | intermediate | 2013 |
| 3. | low | low | intermediate | 2008 |
| 4. | low | high | highest | 2011 |