# Peer review of "Persistence of upper stratospheric winter time tracer variability into the Arctic spring and summer"

_Atmospheric Chemistry and Physics, 2015_

## Referee Comment (RC1) · Anonymous Referee #2 · 10 Apr 2016

This paper contains an analysis of measurements and model simulations of methane (CH4) and chlorine monoxide (ClO) in the upper stratosphere at northern high latitudes from late winter through the early fall seasons. It is shown that interannual variability in winter/spring dynamics leads to large changes in the mean CH4, which persists into the summer, and impacts the level of summertime ClO and ozone (O3). Overall, this is a very significant result as it shows that high latitude winter/spring dynamics in the upper stratosphere and mesosphere has an important effect on summertime O3 in the upper stratosphere through chlorine chemistry, which is distinct from the well-established link to ozone through the descent of nitrogen compounds.

The manuscript is fairly clear overall; the organization of topics, the text, and figures are

all in good shape for publication. However, there are a few critical issues that should be addressed by the authors in order to make this story complete. I recommend that the paper should be published in ACP, provided that the two major issues below can be adequately addressed.

Major

1. Possible effects from sampling biases of SOFIE observations: The manuscript makes it clear that the sampling latitude for SOFIE varies with season between 65oN and 82oN. This is highly appropriate to point out to the reader, but it leads to a number of possible issues that may impact the analysis. Perhaps these issues are addressed in detail by previous publications. If so, then a summary discussion should be included here (with references). If not, then this paper should explore the following impacts in greater detail:

(i) Interpretation of "zonal means" in time-pressure coordinates (Figs 1 and 2) is clouded by the mixing of time and latitude dimensions. Inferring the amount of descent from Fig 1 (discussion lines 115-148) and the impact of photochemistry on the summertime decrease in CH4 (lines 136-139) can only be done if latitudinal sampling effects can be ignored. If the observation latitude is changing by ∼6deg every month, then a month-to-month variation is likely induced through the sampling of background latitudinal gradients in CH4. For example, it is possible that a large fraction of the CH4 decrease seen in every year from day 172 to day 264 in Fig 2 is a result of the latitude decreasing from 56 to 82deg latitude (and similarly for the abrupt increase in CH4 after day 264).

(ii) The number of longitude samples per zonal mean, and their distribution in longitude for the daily (Fig 1) and five-day (Fig 2) means should be reported, in order to address the question of whether these data truly represent zonal means.

(iii) The possible use of equivalent latitude coordinates should be explored. It has been well established that for long-lived tracers such as CH4, time variations in zonal

means can be altered by non-zonal circulation patterns. It is a particularly pronounced effect for zonal means taken near the winter/spring vortex edge (this impacts up to day ~100 in Figs 1, 2, and Fig 5 for O3). Equivalent latitude transformation is probably not possible with SOFIE measurements, but it can and should be done here for MLS ClO observations and also for the WACCM model results. It should be clearly demonstrated that either (i) the impact of combining different dynamical regimes (e.g. inside/outside vortex) has little impact on the CH4, ClO, and O3 variability in zonal means, or (ii) even a large impact in winter/spring does not affect the main conclusion for summertime ozone.

2. A more complete picture of the chemistry, including other compounds, is needed. One of the main conclusions of paper draws on photochemical links, from CH4 to ClO, and from ClO to O3. The description of this chemistry is lacking in detail and leaves important questions unanswered. At a minimum, some interested readers will not be experts in stratospheric/mesospheric chemistry so that a discussion of the issues below is warranted.

(i) The reaction Cl+CH4 -> HCl+CH3 is identified on line 160 as being the reason that ClO varies inversely with CH4 as in Figures 3 and 4. The changes are not 1:1 (hundreds of ppbv for CH4, tenths of ppbv for ClO), but is this expected? Is it a result of other loss processes for CH4? Are the observed slopes of ClO vs CH4 consistent with expectations from chemistry? For example, if there is a true inverse relationship ($y=1/x$, and not $y=1/x2$ or $y=1/sqrt(x)$), then we might expect $dy/y=-dx/x$. Is this the case here? If a causal link is being argued, then it is important to build the foundation by considering and discussing the important chemical processes impacting both CH4 and ClO, such as major production and loss mechanisms.

(ii) Connected to the above questions, it is puzzling why there are no MLS HCl measurements included in the analysis, considering that it is a product of the Cl+CH4 reaction. Are the effects from CH4 variability seen in HCl, and if not, why? A positive results would clear bolster the discussion related to (i) above.

(iii) Diurnal variations. I believe that there may be significant diurnal variability in both ClO and O3 in the upper stratosphere and stratopause regions. Is this an issue for sampling MLS at the SOFIE latitudes? Since MLS is sun-synchronous, then the local time of the sampled MLS measurements will vary with season.

(iv) A clearer picture of the changes in ozone attributed to ClOx chemistry is needed. Although Fig 6 from WACCM/NOGAPS is useful, interpretation of model results is sometimes just as complicated as observations unless there is an accompanying, simplified analytic/theoretical basis. In particular, on lines 208-220 it is stated that ozone loss from chlorine is about 20% larger in 2009 than in 2008. Yet, the net effect considering ClOx, NOx, and HOx loss is only 2% larger in 2009 vs 2008. Is this just an additive sum of the loss rates? Why is NOx lower in 2009? Large descent in that year should have led to higher NOx. How does the WACCM NOx for 2008/2009 compare with measurements, i.e., can we really believe the NOx behavior? (Important - as it offsets most of the ClOx effect.) It also seems odd that the HOx cycle is relatively unchanged between 2008 and 2009; changes in descent likely changed upper stratospheric water vapor (measured by MLS), which should impact HOx. Although it's true that NOx and HOx are not the main foci of this paper, any conclusions involving ClOx - O3 links based on observations should carefully consider the impacts of these other chemical families. This would include both their influence on O3, and possible interrelationships between HOx, NOx, and ClOx.

Minor things to consider:

1. Lines 54-60 and Table 1: Two things to consider here. Since their are 4 combinations to two quasi-binary states, is there any coupling between the two states, i.e., are winter descent and spring PW independent? Also, there is no clear statement of the criteria used to gauge both descent and PW activity. This would include both the quantity used in each case (e.g., slope of tracer contours for descent, or heat fluxes for PW) and the corresponding thresholds for distinguishing "high" from "low".

2. Line 104: should probably define "MERRA"

3. Figure 1: As noted on lines 113-117, this figure illustrates variations in upper strato-spheric CH4 that can be driven by variations in descent from the mesosphere . In looking a cross-stratopause transport, it would be useful to show the stratospause as dashed curves in Fig 1. Since SOFIE measures T, this should be a straightforward addition.

4. Lines 146-148: Contrary to the statement that "once the relative abundance of CH4 was established by May, it remained mostly unchanged until October", it appears from Figure 2 that in 2011, 2014, and 2013, CH4 decreased by around 60%. It is harder to judge some of the other years, but they appear to be around 50%. On the other hand, is this statement mean to convey that all of the years seem to decrease by the same fraction?

5. Figures 5 and 6: The choice of using 80N for looking at ozone and ozone loss rates is not well justified. On the one hand, Figure 4 states that the mean latitude of SOFIE for August is 78deg (and 78deg is used also used in Fig 8 for ozone). On the other hand, if one is trying to quantify the impact on ozone from ClO variations observed over the months of May-August (and sampled at the SOFIE latitudes as in Figure 3), then the mean latitude might be closer to 70deg .

6. Lines 219-220: In Figure 6, at 4 hPa ozone loss is about 10% less in 2009 compared with 2008, whereas at 1.47 hPa it is about 2% larger in 2009 compared with 2008. This is stated as the reason why, in Figure 7, 2009 minus 2008 ozone is positive at 4 hPa and negative at 1.47 hPa. Yet, the 4 hPa change in loss is 5 times larger than at 1.47 hPa whereas the ozone response is smaller at 4hPa than at 1.47 hPa. This is not explained. Is the ozone response nonlinear?

7. Figure 1 caption: "This latitude varies has some variation..."

8. Figure 7: "monthly and daily averaged" is unclear.

---

## Referee Comment (RC2) · Anonymous Referee #2 · 10 Apr 2016

Major concern #1, part (i), last sentence: "decreasing from 56 to 82deg latitude" should read "increasing from 65 to 82deg latitude"

---

## Referee Comment (RC3) · Anonymous Referee #1 · 18 Apr 2016

In this manuscript authors use SOFIE CH4 and MLS (ClO and Ozone) data to show persistence in the upper stratospheric CH4 that is linked to mesospheric downward descent and horizontal transport in the upper stratosphere. Authors also quantify its effect on the upper stratospheric ozone as ozone loss via ClOx cycle is also modified by the background CH4 concentrations. Then authors try to present 4 categories of this transport (a) weak wintertime downward descent, weak horizontal transport, (b) same as (a) but strong horizontal transport, (c) strong wintertime descent but weak mid-to-high latitude horizontal transport and (d) same as (c) but strong horizontal transport. They clearly show strong relationship between SOFIE CH4 and MLS ClO as well as MLS O3 and SOFIE CH4. These results are supported by chemistry climate model

WACCM nudged with NOGAPS-ALPHA data. Overall this is well written scientific manuscript that helps to improve our understanding about upper stratospheric/lower mesospheric chemistry and should be published in ACP, if authors can address some the comments listed below.

Major Comments: (a) I think authors have good understanding about various mesospheric/stratospheric dynamics but that is not well explained in the manuscript. Adding extra paragraph explaining differences between stratospheric and mesospheric dynamics should be included. This would help non-expert reader to get better understanding about background dynamics. My understanding is that mesospheric dynamics primarily driven by breaking of gravity waves that can propagate at higher altitudes whereas long planetary waves break at lower altitudes thereby driving stratospheric circulation (although nothing is completely decoupled, and both the circulation have horizontal and vertical component). But annual cycle in mesospheric and stratospheric circulation is out of phase. Also instead of descent, try to say "mesospheric descent". And when you are discussing "spring-time wave activity", I think you meant to say horizontal transport from low-to-high latitudes in the stratosphere.

(b) Main conclusions are primarily based on SOFIE data but it's description is very short. These profiles must have been some validation against other instruments. Please include some references. Also give some information about the retrieval and instrument errors.

Minor comments: i) line 5: How do define "descent" and "spring-time wave activity". These two are not exactly opposite terms.

ii) Line 28: it's not just propagation; it is "breaking of waves" that is more important.

iii) Line 81: What about rewording it as "Validation of MLS ClO against . . .. Is presented in Santee et al.,

iv) Line 90: Is it simple linear interpolation or you use information form SOFIE averaging

kernel (which should be scientifically correct method).

v) Can you comment on problems in MERRA data

vi) Figure 1: Why 2012/13 is not there?

vii) Lines 160-165: What altitudes?

viii) Line 166: Also give altitude or 1.47 hPa

ix) Minor suggestion: Why you can't show HCl?. That would have help you present complete story. Do you think any sudden changes HCl trend in the NH can be explained with changes in CH4 transport (Mahieu et al., 2014, Nature).

---

## Author Comment (AC1) · 5 May 2016

We are pleased to have the opportunity to answer the referees' questions and thank them for their comments. Below we propose some changes and some new figures, all of which we believe add to the robustness and clarity of our results. The original comments from the reviewer are in blue italics.

Referee 1

*Major Comments:*

*(a) I think authors have good understanding about various mesospheric/ stratospheric dynamics but that is not well explained in the manuscript. Adding extra paragraph explaining differences between stratospheric and mesospheric dynamics should be included. This would help non-expert reader to get better understanding about background dynamics. My understanding is that mesospheric dynamics primarily driven by breaking of gravity waves that can propagate at higher altitudes whereas long planetary waves break at lower altitudes thereby driving stratospheric circulation*
*(although nothing is completely decoupled, and both the circulation have horizontal and vertical component). But annual cycle in mesospheric and stratospheric circulation is out of phase. Also instead of descent, try to say "mesospheric descent". And when you are discussing "spring-time wave activity", I think you meant to say horizontal transport from low-to-high latitudes in the stratosphere.*

We can add some references in the Introduction, specifically to Siskind et al., JGR, 2010, "Case studies of the mesospheric response to minor, major and extended stratospheric warmings" and to Chandran et al., JGR, 2013, "A climatology of elevated stratopause events in the whole atmosphere community climate mode"; We will insure that that we add the word mesospheric whenever descent is discussed- it already is in many places, but not yet at the end of Section 3.0.1. We will also reference Siskind et al., 2015a who discuss how spring time wave activity is associated with horizontal transport of low latitude air to polar latitudes.

*(b) Main conclusions are primarily based on SOFIE data but it's description is very short. These profiles must have been some validation against other instruments. Please include some references. Also give some information about the retrieval and instrument errors.*

As far as SOFIE $CH_4$, it is true that a detailed validation of $CH_4$ has not yet been published. The best we can say is that in studies of wintertime mesospheric descent (Bailey et al., 2014; Siskind et al., 2015b) it appears to display consistent features when compared with other SOFIE tracers. We thus added mention of that. We also added the precision estimate of the $CH_4$ data from Gordley et al. (2009) (10 ppbv at 70 km).

*Minor comments:*

*i) line 5: How do define "descent" and "spring-time wave activity". These two are not exactly opposite terms.*

Line 5: Actually, it could be argued that these are opposites. As noted above, springtime PW activity has been linked to cooling of the mesosphere (cf. Siskind et al., 2015a). GW driven mesospheric descent leads to warming of the mesosphere (Siskind et al., 2010; Chandran et al., 2013).

*ii) Line 28: it's not just propagation; it is "breaking of waves" that is more important.*
Actually, the reviewer's interpretation is not necessarily correct. As has been shown (references above), the change in the background zonal winds changes the propagation of gravity waves with different phase speeds. Certain waves that were filtered out can now propagate up to the upper mesosphere. I would argue that propagation is the correct word.

*iii) Line 81: What about rewording it as "Validation of MLS ClO against : : :. Is presented in Santee et al.,*

We do not understand the reason for the suggested reword and feel the text as written accurately describes the literature. Santee wrote a validation paper which was more than simply comparing with other datasets; it was a complete error analysis. Nedoluha compared MLS with ground based data.

*iv) Line 90: Is it simple linear interpolation or you use information form SOFIE averaging kernel (which should be scientifically correct method).*

It is simple linear interpolation. We do not feel it is correct to use SOFIE $CH_4$ retrieval kernels in a discussion of MLS ClO.

*v) Can you comment on problems in MERRA data*

We do not understand the question about "problems with MERRA". We do not think we imply that there are any problems other than it lacks a mesosphere. That is not relevant for the present paper but is discussed a bit more in Siskind et al., 2015b)

*vi) Figure 1: Why 2012/13 is not there?*
We do not understand the question. 2012 and 2013 are both shown and labeled (at the bottom)

upper stratosphere for Siskind et al., 1998 and 2 hPa for Froidevaux et al,. 2000

*viii) Line 166: Also give altitude or 1.47 hPa*
Agree, we will add "1.47 hPa"

*Minor suggestion: Why you can't show HCl?. That would have help you present*
*complete story. Do you think any sudden changes HCl trend in the NH can be explained*
*with changes in CH4 transport (Mahieu et al., 2014, Nature).*

As noted below in our response to Reviewer #2, according to the Aura web site, there is no HCl
data above 30 km after 2006. Indeed, the Mahieu paper cited above explicitly states that the top
boundary of their GOZCARDS merged data product is 31 km.

Referee 2

*Major*
*1. Possible effects from sampling biases of SOFIE observations: The manuscript*
*makes it clear that the sampling latitude for SOFIE varies with season between 65$^o$N*
*and 82$^o$N. This is highly appropriate to point out to the reader, but it leads to a number*
*of possible issues that may impact the analysis. Perhaps these issues are addressed*
*in detail by previous publications. If so, then a summary discussion should be included*
*here (with references). If not, then this paper should explore the following impacts in*
*greater detail:*

*(i) Interpretation of "zonal means" in time-pressure coordinates (Figs 1 and 2) is*
*clouded by the mixing of time and latitude dimensions. Inferring the amount of descent*
*from Fig 1 (discussion lines 115-148) and the impact of photochemistry on the*
*summertime decrease in CH4 (lines 136-139) can only be done if latitudinal sampling*
*effects can be ignored. If the observation latitude is changing by _6deg every month,*
*then a month-to-month variation is likely induced through the sampling of background*
*latitudinal gradients in CH4. For example, it is possible that a large fraction of the CH4*
*decrease seen in every year from day 172 to day 264 in Fig 2 is a result of the latitude*
*increasing from 56 to 82deg latitude (and similarly for the abrupt increase in CH4 after*
*day 264).*
*(ii) The number of longitude samples per zonal mean, and their distribution in longitude*
*for the daily (Fig 1) and five-day (Fig 2) means should be reported, in order to address*
*the question of whether these data truly represent zonal means.*
*(iii) The possible use of equivalent latitude coordinates should be explored. It has*
*been well established that for long-lived tracers such as CH4, time variations in zonal* means can
*be altered by non-zonal circulation patterns. It is a particularly pronounced*

*effect for zonal means taken near the winter/spring vortex edge (this impacts up to day
_100 in Figs 1, 2, and Fig 5 for O3). Equivalent latitude transformation is probably not
possible with SOFIE measurements, but it can and should be done here for MLS ClO
observations and also for theWACCM model results. It should be clearly demonstrated
that either (i) the impact of combining different dynamical regimes (e.g. inside/outside
vortex) has little impact on the CH4, ClO, and O3 variability in zonal means, or (ii) even
a large impact in winter/spring does not affect the main conclusion for summertime
ozone.*

Concerning the range of latitudes and the sampling. We should first note that this paper deals with mostly summer conditions- the polar vortex is anticyclonic and generally stable. Thus the sampling problems associated with winter time conditions are greatly, if not entirely, reduced. It is, however, useful to more clearly show that the drift in SOFIE latitudes does not factor into our interpretation. To that end, we have added Figure 2b. Figure 2b is in the same format as Figure 2a and shows 2008 and 2009 WACCM monthly averaged CH4, deliberately sampled at a single latitude, 75°N. The agreement with the 2008 and 2009 curves shown in Figure 2a is excellent (after the late winter), proving that the slow change in SOFIE latitudes is not misleading us.

[Figure]

**Figure 2. Time series of zonal mean CH4 mixing ratio at 1.47 hPa (a) SOFIE data for the indicated years. The data have been grouped in 5-day bins. See Figure 1 for a discussion of the latitudes. (b) WACCM for 2008 (solid) and 2009 (dashed) at a single fixed latitude of 75°N.**

SOFIE acquires approximately 15 samples/day, uniformly spaced in longitude.  This will be stated in the revised copy.

*A more complete picture of the chemistry, including other compounds, is needed.
One of the main conclusions of paper draws on photochemical links, from CH4 to ClO,
and from ClO to O3. The description of this chemistry is lacking in detail and leaves
important questions unanswered. At a minimum, some interested readers will not be
experts in stratospheric/mesospheric chemistry so that a discussion of the issues below*

*is warranted.*

*(i) The reaction Cl+CH4 -> HCl+CH3 is identified on line 160 as being the reason that ClO varies inversely with CH4 as in Figures 3 and 4. The changes are not 1:1 (hundreds of ppbv for CH4, tenths of ppbv for ClO), but is this expected? Is it a result of other loss processes for CH4? Are the observed slopes of ClO vs CH4 consistent with expectations from chemistry? For example, if there is a true inverse relationship (y=1/x, and not y=1/x2 or y=1/sqrt(x)), then we might expect dy/y=-dx/x. Is this the case here? If a causal link is being argued, then it is important to build the foundation by considering and discussing the important chemical processes impacting both CH4 and ClO, such as major production and loss mechanisms.*

Regarding the details of the relationship between ClO and CH4, responding to this question proved interesting. Froidevaux et al [2000] present an explanation of the ClO and CH4 relationship in the upper stratosphere and we will cite that. One consequence of their formulation of possible inverse power relationships is that the slope should be greater for lower values of CH4. Our results show precisely this, a steepening of the slope (more negative) as the summer progresses. The specific values for this slope are now shown as "m" in each panel of Figure 3. Further, for higher values of CH4, seen in the tropics, Nedoluha et al [2011] quote a lower value for the ClO/CH4 slope, as expected. Thus all three papers, ours, Nedoluha et al and Froidevaux are all quite consistent and we believe this adds to the credibility and robustness of our results.

[Figure]

**Figure 3. Scatterplot of zonal, mean averaged MLS ClO versus SOFIE CH4 at 1.47 hPa. The MLS data are sampled at the SOFIE occultation latitude, the monthly averages of which are indicated in each panel. In the upper right of each panel are given the linear correlation coefficients (r) between each dataset for each month and the slope of the linear fit (m) in units of ppbv of ClO per ppmv of CH$_4$.**

Having said all that, we prefer not to get into the details of the precise functional form of the relationship which is some of what the reviewer is asking about. All we are pointing out is that the slope is different when you consider a wide range of CH4 values. And that the change is of the correct sense as expectations and places our work in a consistent context with Froidevaux and Nedoluha. The consistency between Nedoluha's analysis and Froidevaux's was not obvious

until our result. It might be interesting for future work to try and narrow down the precise relationship but that is beyond the scope of the present study.

*(ii) Connected to the above questions, it is puzzling why there are no MLS HCl measurements included in the analysis, considering that it is a product of the Cl+CH4 reaction.*
*Are the effects from CH4 variability seen in HCl, and if not, why? A positive results would clear bolster the discussion related to (i) above.*

We did not use MLS HCl, because on the Aura/MLS/HCl web page they clearly state that the instrument degraded after 2006 such that there is essentially no HCl data above 30 km.

*(iii) Diurnal variations. I believe that there may be significant diurnal variability in both ClO and O3 in the upper stratosphere and stratopause regions. Is this an issue for sampling MLS at the SOFIE latitudes? Since MLS is sun-synchronous, then the local time of the sampled MLS measurements will vary with season.*

There is likely no diurnal variation in ClO because there is no nighttime. For latitudes poleward of 65 degrees, from April to September, MLS is in almost continuous sunlight. This is already noted in the original text (line 90).

*(iv) A clearer picture of the changes in ozone attributed to ClOx chemistry is needed. Although Fig 6 from WACCM/NOGAPS is useful, interpretation of model results is sometimes just as complicated as observations unless there is an accompanying, simplified analytic/theoretical basis. In particular, on lines 208-220 it is stated that ozone loss from chlorine is about 20% larger in 2009 than in 2008. Yet, the net effect considering ClOx, NOx, and HOx loss is only 2% larger in 2009 vs 2008. Is this just an additive sum of the loss rates? Why is NOx lower in 2009? Large descent in that year should have led to higher NOx. How does the WACCM NOx for 2008/2009 compare with measurements, i.e., can we really believe the NOx behavior? (Important - as it offsets most of the ClOx effect.) It also seems odd that the HOx cycle is relatively unchanged between 2008 and 2009; changes in descent likely changed upper stratospheric water vapor (measured by MLS), which should impact HOx. Although it's true that NOx and HOx are not the main foci of this paper, any conclusions involving ClOx - O3 links based on observations should carefully consider the impacts of these other chemical families. This would include both their influence on O3, and possible interrelationships between HOx, NOx, and ClOx.*

Regarding the overall ozone balance, we have to clarify that Figure 6 is not explained very well. First there is a mistake on our part. Figure 6 shows not the ozone loss rates, but the total odd oxygen loss rates. These rates are a function of odd oxygen itself (cf. equation A1 of McCormack et al., 2006), and the odd oxygen abundance is different from 2008 to 2009. Thus describing Figure 6 has to be done more carefully than we did in the original manuscript. One

cannot simply take the difference between the 2008 and 2009 odd oxygen loss rates and precisely map them to ozone changes. The original text is misleading in this regard and we will rewrite it to clarify that that is simply shows the contribution to total odd oxygen loss from the different cycles. Our apologies for our earlier oversimplification.

Regarding NOx, we agree that documenting its variation is useful to provide context to the observed ClO and $O_3$ changes. SOFIE does not measure the $NO_2$ component of NOx so we show, as the new Figure 7, the 2009 and 2008 changes from WACCM. As has been clearly documented in Siskind et al 2015b, the WACCM/NOGAPS NOx is likely an excellent representation of reality. The figure clearly shows the offsetting effect of the lower NOx in the upper stratosphere in 2009. At pressures greater than 3 hPa, where Figure 6 shows the NOx cycle dominates, this explains the ozone increase in 2009 relative to 2008. Further, as it already states in the text, there is no evidence that the NOx enhancements seen in 2009 ever penetrated down to altitudes to where the NOx catalytic cycle effects ozone (Siskind et al, 2015b and Salmi et al., 2011). We suggest that the lower NOx from 1-8 hPa in 2009 seen in the figure is a legacy of greater winter/spring descent from the region of the NO minimum in the mesosphere near 60-75 km.

[Figure]

**Figure 7. Monthly averaged WACCM/NOGAPS NOx (=NO + NO2) for June 2009 (solid) and 2008 (dashed) at 75°N.**

*Minor things to consider:*

*1. Lines 54-60 and Table 1: Two things to consider here. Since their are 4 combinations to two quasi-binary states, is there any coupling between the two states, i.e., are winter descent and spring PW independent? Also, there is no clear statement of the criteria used to gauge both descent and PW activity. This would include both the quantity used in each case (e.g., slope of tracer contours for descent, or heat fluxes for PW) and the corresponding thresholds for distinguishing "high" from "low".*

The distinction made in Table 1 is qualitative and essentially relative over the 6 years. Butler et al recently discussed in the November 2015 issue of the Bulletin of the American Meteorological Society that after over 50 years, there is still no quantitative consensus on what a sudden stratospheric warming is. Similar ambiguities exist for with more specialized topics such as elevated stratopauses, FrIACs etc. Given these level of uncertainties about such wide range of processes, it seems unnecessary to try and give a more quantitative definition to the terms used in Table 1. We propose to add an additional sentence stating that. Perhaps in the future, with sufficient CH4 data, it would be possible to use the CH4 mixing ratio at certain altitudes and certain times of year as more quantitative measures.

*2. Line 104: should probably define "MERRA"*

Concur. We had actually defined it on our $2^{nd}$ use, which admittedly is confusing and we apologize for that.

*Figure 1: As noted on lines 113-117, this figure illustrates variations in upper stratospheric CH4 that can be driven by variations in descent from the mesosphere . In looking a cross-stratopause transport, it would be useful to show the stratospause as dashed curves in Fig 1. Since SOFIE measures T, this should be a straightforward addition.*

The problem with this is that the definition of stratopause is variable- the temperature maximum is up near 80 km at the beginning of the time series (and off scale).

*Lines 146-148: Contrary to the statement that "once the relative abundance of CH4 was established by May, it remained mostly unchanged until October", it appears from Figure 2 that in 2011, 2014, and 2013, CH4 decreased by around 60%. It is harder to judge some of the other years, but they appear to be around 50%. On the other hand, is this statement mean to convey that all of the years seem to decrease by the same fraction?*

While the absolute abundance of CH4 does indeed decrease, the sentence does not say that. It says the relative abundance- meaning from one year related to the others to the next, remained unchanged. We can however, certainly reword this sentence slightly to try and make that clearer.

*Figures 5 and 6: The choice of using 80N for looking at ozone and ozone loss rates is not well justified. On the one hand, Figure 4 states that the mean latitude of SOFIE for August is 78deg (and 78deg is used also used in Fig 8 for ozone). On the other hand, if one is trying to quantify*

*the impact on ozone from ClO variations observed over the months of May-August (and sampled at the SOFIE latitudes as in Figure 3), then the mean latitude might be closer to 70deg .*

We will now use 75N, consistent with the new Figure 2b. We also state now that they're averages for the month of June 2009 and 2008. An example of this is the new figure 7 shown above.

*Lines 219-220: In Figure 6, at 4 hPa ozone loss is about 10% less in 2009 compared with 2008, whereas at 1.47 hPa it is about 2% larger in 2009 compared with 2008. This is stated as the reason why, in Figure 7, 2009 minus 2008 ozone is positive at 4 hPa and negative at 1.47 hPa. Yet, the 4 hPa change in loss is 5 times larger than at 1.47 hPa whereas the ozone response is smaller at 4hPa than at 1.47 hPa. This is not explained. Is the ozone response nonlinear?*

We feel this is addressed above in our discussion of the chemistry and the inclusion of the new figure showing the NOx variation from WACCM

On the figure captions, we concur. This will be corrected (should just say "monthly averaged")